# Studying dynamics in two-dimensional quantum lattices using tree tensor network states

Benedikt Kloss[1][*], David R. Reichman[1][†] and Yevgeny Bar Lev[2][‡]

**1** Department of Chemistry, Columbia University,
3000 Broadway, New York, New York 10027, USA
**2** Department of Physics, Ben-Gurion University of the Negev, Beer-Sheva 84105, Israel

[*] bk2576@columbia.edu † drr2103@columbia.edu ‡ ybarlev@bgu.ac.il

## Abstract

We analyze and discuss convergence properties of a numerically exact algorithm tailored to study the dynamics of interacting two-dimensional lattice systems. The method is based on the application of the time-dependent variational principle in a manifold of binary and quaternary Tree Tensor Network States. The approach is found to be competitive with existing matrix product state approaches. We discuss issues related to the convergence of the method, which could be relevant to a broader set of numerical techniques used for the study of two-dimensional systems.

 Check for updates

## Contents

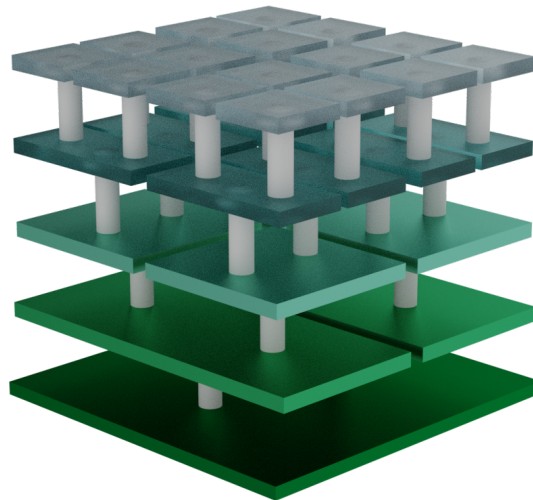

Figure 1: Illustration of binary TTNS structure for a 4x4 lattice. The physical degrees of freedom are on the topmost layer and the top node is in the bottom-layer of the figure.

# 1 Introduction

The exact simulation of the non-equilibrium dynamics of interacting quantum lattice systems is generally an unsolved challenge, due to the exponential growth of the Hilbert space with the size of the system. Tensor network state (TNS) methods allow for a significant extension of accessible length scales by trading in the exponential cost in system size for an exponential cost in time. This becomes possible due to a reduction of the exact Hilbert space in terms of a structured product of low-order tensors, referred to as a tensor network. The set of the states expressible by a given tensor network spans only a small region in the full Hilbert space, but the coverage can be improved systematically by increasing the number of variational parameters, i.e. the bond dimension. For partitions of the lattice that lead to simply-linked tensor network parts, the logarithm of the bond-dimension gives an upper bound to the entanglement entropy. Since the entanglement of a generic system after a global quench grows linearly with time [1–3], the accessible timescales are limited. In one-dimensional systems, these timescales are often comparable to those attainable in experimental realizations [4], however going to higher spatial dimensions becomes extremely challenging due to a number of reasons.

While in one-dimensional systems, matrix product states (MPS) are known to efficiently represent area-law entangled states (which includes ground-states of gapped one-dimensional systems), this does not hold in two spatial dimensions [5–7]. The generalization of MPS to two-dimensional lattices is called Projector-Entangled Pair States (PEPS) [8], which provides an efficient representation of two-dimensional area-law entangled states [9], but PEPS are challenging to manipulate numerically [10] (see also Ref. [11] for a recent review). Approximations that are hard to control are typically used in PEPS algorithms in order to tame the computational effort. Even with such approximations, the computational scaling is usually unfavorably steep. Nonetheless, PEPS-derived methods are state-of-the art numerical techniques for computing ground-states of two-dimensional systems [12]. Extensions of PEPS methods to the time-domain have been recently developed, however the accessible timescales are extremely limited [13–16]. An alternative approach is to use tensor network structures, which are more numerically tractable. One way to achieve this is to map the two-dimensional lattice into a one-dimensional chain and apply MPS methods, which are adjusted to handle the long-ranged interactions that arise from the mapping [17–22]. Ref. [20], for example,

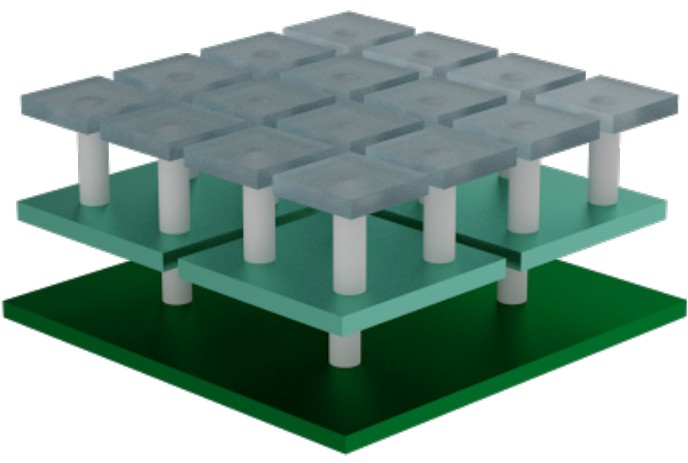

Figure 2: Same as Fig. 1 but for a quaternary TTNS.

introduced an algorithm which expresses the propagator as a matrix product operator (MPO) acting on the states encoded as MPS. The application of this approach to two-dimensional lattices shares the very limited timescale of the more recent approaches based on PEPS, since the advantages in the computational scaling of simpler tensor networks are balancing out the disadvantages in non-optimal representation of entanglement by the tensor network structure for the problem at hand. A novel development is the use of artificial neural networks (ANN) to encode the wavefunction and its time-evolution [23]. They have been shown to perform competitively with state-of-the art TNS techniques in recent applications to two-dimensional systems [24, 25]. However, much remains to be learned about the possibilities and limitations of such methods.

It is important to explore computationally tractable tensor network structures other than MPS, since they may enable progress in the computation of the exact dynamics of interacting two-dimensional systems. For this purpose, in this work we propose to employ Tree Tensor Network States (TTNS), which encompass all loop-free tensor network states. While similarly to MPS, hierarchical, tree-like TTNS can only efficiently encode states with area-law entanglement in one dimensional systems they offer a more robust description of ground states of *critical* one dimensional systems [26, 27], and therefore might provide more flexibility in encoding complex entanglement structures in two and higher dimensional systems. TTNS are used in the context of interacting lattice systems [28–36], but they feature more prominently in applications like electronic structure methods [37–39] or molecular quantum dynamics in the chemical physics literature. In this context they are called the Multi-Configuration Time-Dependent Hartree (MCTDH) method and its multi-layer generalization (ML-MCTDH) [40–42]. In ML-MCTDH, the time-dependent variational principle [43, 44] (TDVP) is applied to a TTNS as a variational ansatz for the wavefunction. Up to differences in the numerical integrations scheme, these methods are similar to the more recent applications of the TDVP tailored specifically to matrix product states [45–49].

The TDVP applied to TNS has been discussed as a method that may enable the accurate description of hydrodynamic transport in non-integrable systems when used with a moderate bond dimension [50], but was shown to not be a robust approximation for generic systems [51]. Several tensor network techniques have been designed to circumvent the entanglement growth on intermediate timescales with the goal of a reliable approximation to the long-time dynamics [52–58]. Despite promising results, the stability of such approximations for generic systems, especially beyond one dimension, is not sufficiently established at this point. In this

work, we thus consider the TDVP applied to TNS as a numerically exact technique, allowing to compute the dynamics of a system within a controllable accuracy up to some finite time, and generalize the algorithms of Refs. [48, 49, 59] to general TTNS. We note in passing that such algorithms have been used to find the ground state of a two-dimensional spin system [60] and to obtain the dynamics of a zero-dimensional model [61]. Recently, similar versions of this algorithm were reported in detail in Refs. [62, 63], which we became aware of during the preparation of this manuscript. While in our work we focus on two-dimensional systems, Ref. [62] showcases a promising application of a TTNS as an impurity-solver, which is an effectively zero-dimensional problem. On the other hand, Ref. [63] proves the algorithm's exactness property as well as a linear error-bound for the total time evolution in the time-step.

The purpose of this work is to investigate the performance of TTNS as a numerically exact method to study the dynamics of two-dimensional systems. In Sec. 2, we introduce the main concepts of TTNS along with the TDVP before presenting the algorithm and commenting on some caveats which are relevant to the applications of the TDVP. We benchmark the method on an exactly solvable, non-interacting two-dimensional system, and compare our approach to previously published results for two-dimensional interacting hardcore bosons in Sec. 3. Notably, we identify the reachable timescales and investigate convergence properties of the algorithm alongside with practical considerations regarding how to assess the accuracy of the results. We conclude by placing the results in the context of existing techniques and recent developments in Sec. 4.

## 2 Theory

Tensor network states represent a pure state, $|\Psi\rangle = \sum_{s_1 \dots s_N} \Psi_{s_1 \dots s_N} |s_1 \dots s_N\rangle$, of a lattice system as a product of tensors $\{T\}$. Each tensor $T_i$ may have a number of indices corresponding to physical degrees of freedom and also auxiliary indices which do not correspond to physical degrees of freedom. Consider the Schmidt decomposition, corresponding to a bipartition of the lattice into a set of sites $A$ and its complement $B$, $\Psi_{s_1 \dots s_N} = \sum_{i,j} \phi^A_{i s_A} \lambda_{ij} \phi^B_{j s_B}$ with $\lambda_{ij} = \delta_{ij} \lambda_i$. This expression can be also understood as a product of three tensors, where a single auxiliary index of each tensor is shared with the diagonal matrix of the Schmidt coefficients (or singular values) $\lambda_i$. In a general tensor network any auxiliary index will appear on two tensors, and summation over the common index implies contraction of the two tensors. Tensor networks can be represented diagrammatically, see Fig. 3a, where the nodes correspond to tensors and the links, dubbed *legs* in the following, indicate a shared index between the two tensors. Any tensor network for which the legs do not form closed loops is considered a *Tree Tensor Network (TTN)*, with matrix product states (MPS) serving as a prominent special case, which is mostly applicable for one-dimensional lattices. Here, we focus on more general TTNS with a simple hierarchical structure: *n-ary TTNS* in which every node has one *parent* node and *n child* nodes, except for those in the top and bottom layers. We group all physical degrees of freedom into the bottom layer such that all layers above the bottom layer contain only nodes with auxiliary legs (see Figs. 3a)–2 for illustration). Without restricting the generality, in this Section we will limit the discussion to binary TTNS. In such TTNS, a general node represents a third order tensor $\Lambda^{[l,i]}$, where $l$ denotes the layer of the tree to which the node belongs, and $i$ enumerates the nodes in that layer. Each such node has two child nodes. Due to the lack of loops in the tensor network, the physical degrees of freedom separate naturally into two groups from the perspective of a node $\Lambda^{[l,i]}$: those reachable by only descending in the tree towards the bottom layer, i.e. those in the subtree of $\Lambda^{[l,i]}$, and their complement. We define the number of non-zero singular values of the Schmidt decomposition along this bipartition as the rank $r$ of node $\Lambda^{[l,i]}$. For a state with volume law entanglement, the exact rank $r$ will

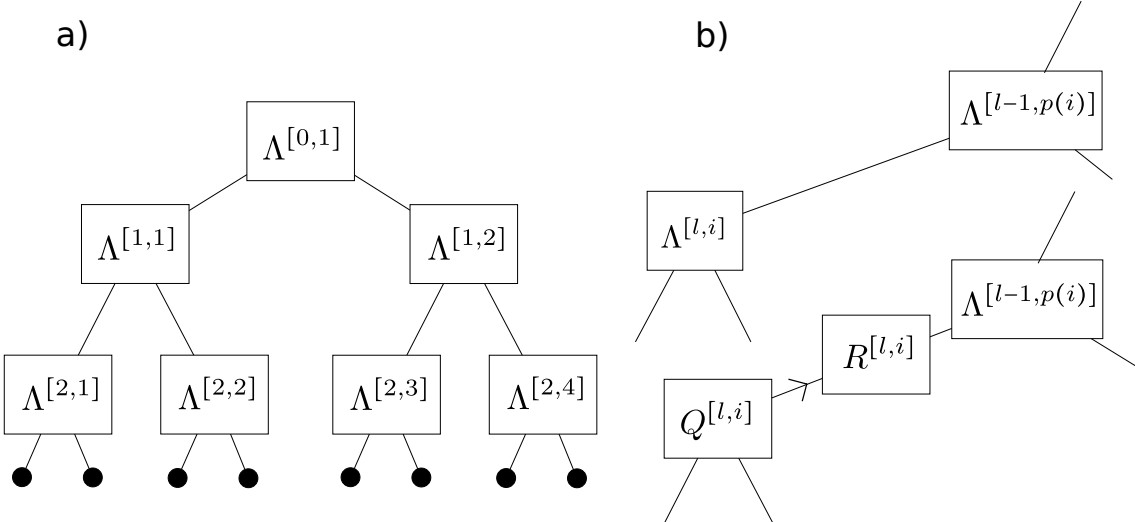

Figure 3: a) A binary TTNS for an 8-site system. The black dots correspond to physical sites and the square boxes with $n$ legs represent $n$-th order tensors. b) Application of the QR decomposition to tensors in the TTNS. The upper and lower diagram represent the same TTNS. The arrow on the link indicates the direction along which the tensor is orthonormalized.

generally scale exponentially with the system size. Thus we introduce a cutoff in the number of kept singular values, namely the bond dimension of the tree $\chi$. In the following, we consider a TTNS of rank $\chi$, which implies that all its tensors $\Lambda^{[l,i]}$ have a rank of $\min\left(\chi, d^{N(l,i)}\right)$, where $d$ denotes the local Hilbert space dimension and $N(l,i)$ is the number of sites in the subtree of $\Lambda^{[l,i]}$. The set of TTNS with a given rank $\chi$ constitutes a smooth manifold of states $\mathcal{M}_\chi$. The computational complexity for a binary TTNS is $\mathcal{O}\left(N \log N \chi^3\right)$ in memory and $\mathcal{O}\left(N \log N \chi^4\right)$ in computation where $N$ is the number of physical degrees of freedom.

We next present a method for time-propagation on the manifold $\mathcal{M}_\chi$ of tree tensor networks with tree rank $\chi$ using a time-dependent variational principle (TDVP) [43, 44]. We start by introducing a few properties and manipulations of TTNS and then describe TDVP and its application to TTNS. We finally highlight important technical details in the use of the TDVP.

## 2.1 TTNS - Basics

A TTNS of a rank $\chi$ is unique up to unitary transformations. This can be seen by inserting a unit matrix between two linked nodes of the tree

$$\Lambda^{[l,i]}_{\alpha_1\alpha_2\alpha_3}\mathbb{I}_{\alpha_3\beta_1}\Lambda^{[l+1,j]}_{\beta_1\beta_2\beta_3} = \Lambda^{[l,i]}_{\alpha_1\alpha_2\alpha_3}U^*_{\alpha_3\gamma}U_{\gamma\beta_1}\Lambda^{[l+1,j]}_{\beta_1\beta_2\beta_3} = \tilde{\Lambda}^{[l,i]}_{\alpha_1\alpha_2\gamma}\tilde{\Lambda}^{[l+1,j]}_{\gamma\beta_2\beta_3}, \tag{1}$$

where repeated indices are summed over, $\mathbb{I}$ represents a $\chi \times \chi$ unit matrix and $U^*$ indicates complex conjugation of the corresponding tensor. This property can be exploited to *isometrize* the tree around any of its nodes [32, 64], which is a generalization of the mixed canonical representation of MPS. To illustrate this concept, consider the *isometrization* about the top-node. In this perspective, every tensor in the tree, except the top-node, represents a truncated orthonormal basis in the space of the bases of child nodes, called *isometry* in the language of real-space or tensor RG. Through recursion, a structured, incomplete basis for the physical lattice sites is obtained. The coefficients for these basis functions are contained in the top node. Any general TTNS can be brought into this form using a sequence of QR decompositions. Practically, one applies QR factorization $\Lambda^{[l,i]}_{\alpha_1\alpha_2\alpha_3} = Q^{[l,i]}_{\beta\alpha_2\alpha_3}R^{[l,i]}_{\beta\alpha_1}$ with $Q^{[l,i]*}_{\beta\alpha_2\alpha_3}Q^{[l,i]}_{\gamma\alpha_2\alpha_3} = \delta_{\beta,\gamma}$, for

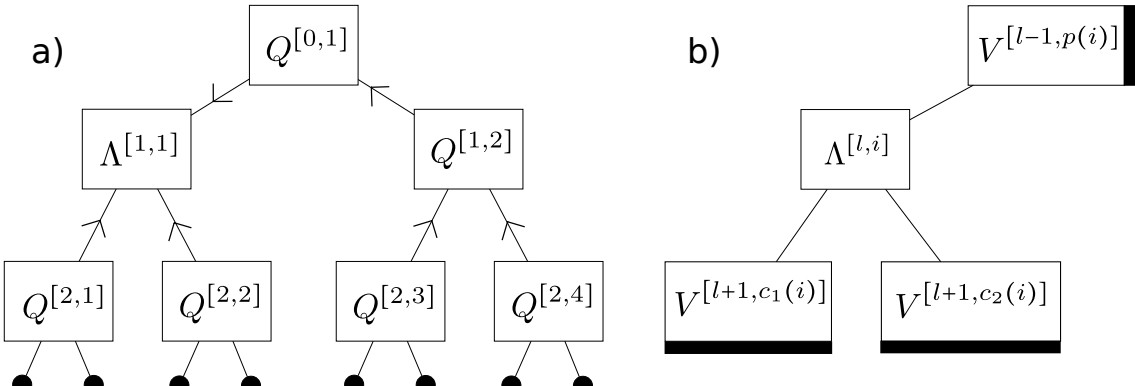

Figure 4: A TTNS isometrized about node [1,1], a), and its shorthand notation, b). The thick black bars on the environment tensors represent the set of physical sites belonging to each of the environment tensors. Note that orthogonality of the environment tensors in b) is not indicated by arrows on the legs, but implicit in their definition.

each of the nodes proceeding layer by layer from bottom to top and absorbing the matrices $R$ into the parent node after each factorization (see also Fig. 3b). Graphically, the direction along which the tensors are orthogonalized is indicated by an arrow on the linking leg. Isometrization around a specific node in the tree translates into arrows pointing in the direction of this node on any (direct) path between the node and a physical site, see Fig. 4b). We may rewrite the TTNS in the following manner:

$$\Psi[l,i]_{\mathbf{s}} = \Lambda^{[l,i]}_{\alpha_1\alpha_2\alpha_3} V^{[l-1,p(i)]}_{\alpha_1\mathbf{s}_1} V^{[l+1,c_1(i)]}_{\alpha_2\mathbf{s}_2} V^{[l+1,c_2(i)]}_{\alpha_3\mathbf{s}_3}. \tag{2}$$

Here, we take the TTNS to be isometrized about node $[l,i]$, indicated as $\Psi[l,i]$, and an environment tensor $V^{[l\pm 1,p(i)/c_j(i)]}_{\alpha_j\mathbf{s}_j}$ is the contraction of all tensors between the legs of node $\Lambda^{[l,i]}$, labeled by $\alpha_j$, and the physical sites $\mathbf{s}_j$, linked by paths from leg $\alpha_j$ that do not cross node $\Lambda^{[l,i]}$. $c_j(i)$ and $p(i)$ are placeholders for the child and parent of node $\Lambda^{[l,i]}$, respectively. We note in passing that similarly to MPS methods, such a contraction is never explicitly carried out, and we only use it for notational convenience. For future reference, we define projectors onto environment tensors of the lower and upper levels in the hierarchy: $\left(\Omega^{[l+1,c_{j-1}(i)]}\right)_{\mathbf{s}'_j\mathbf{s}_j} = V^{[l+1/c_{j-1}(i)]}_{\alpha_j\mathbf{s}'_j} V^{[l+1/c_{j-1}(i)]*}_{\alpha_j\mathbf{s}_j}$ and $\left(\Omega^{[l-1,p(i)]}\right)_{\mathbf{s}'_1\mathbf{s}_1} = V^{[l-1/p(i)]}_{\alpha_1\mathbf{s}'_1} V^{[l-1/p(i)]*}_{\alpha_1\mathbf{s}_1}$. A useful property of the environment tensors is their orthogonality, which allows for efficient calculation of certain physical quantities. For example, if the state is isometrized about node $[l,i]$, the norm of the state is given by $\langle\Psi[l,i]|\Psi[l,i]\rangle = \Lambda^{[l,i]*}_{\alpha_1\alpha_2\alpha_3}\Lambda^{[l,i]}_{\alpha_1\alpha_2\alpha_3}$ since $V^{[l\pm 1,p(i)/c_j(i)]*}_{\alpha_j\mathbf{s}_j} V^{[l\pm 1,p(i)/c_j(i)]}_{\alpha'_j\mathbf{s}_j} = \delta_{\alpha'_j\alpha_j}$. To improve the readability of the presentation, in the following we will omit the indices specifying the elements of the tensors.

## 2.2 TDVP

The time-dependent variational principle generates classical dynamics in the space of variational parameters, $\alpha$, described by the Lagrangian

$$\mathcal{L}[\alpha,\dot\alpha] = \langle\Psi[\alpha]|i\partial_t|\Psi[\alpha]\rangle - \langle\Psi[\alpha]|\hat{H}|\Psi[\alpha]\rangle. \tag{3}$$

The associated action is minimized along a path on a certain variational manifold, which in our case is the manifold of TTNS with tree rank $\chi$, $\mathcal{M}_\chi$. The principle of least-action yields

the following equation of motion,

$$i\partial_t |\Psi[\alpha]\rangle = P_T(\Psi[\alpha])\hat{H} |\Psi[\alpha]\rangle, \tag{4}$$

where $P_T(\Psi[\alpha])$ is the projector onto the tangent space of the manifold $\mathcal{M}_\chi$ at the point $\Psi[\alpha]$. An expression for $P_T(\Psi[\alpha])$ was derived for general binary TTNS in Refs. [65, 66]. Here, we will use an additive splitting of $P_T(\Psi[\alpha])$, in an analogy to those presented for TTNS with only two layers, i.e. Tucker tensors [59] and matrix product states [48, 49], respectively. Note that the latter two TNS are subclasses of a general TTNS and that the expressions for the projector, $P_T(\Psi[\alpha])$, is not restricted to binary TTNS and is valid for *any* TTNS with straightforward modifications. In particular,

$$P_T(\Psi[\alpha]) = P_0 + \sum_{[l,i]} P_+^{[l,i]} - P_-^{[l,i]}, \tag{5}$$

with

$$P_0 = \Omega^{[1,1]}\Omega^{[1,2]} \tag{6}$$
$$P_+^{[l,i]} = \Omega^{[l+1,c_1(i)]}\Omega^{[l+1,c_2(i)]}\Omega^{[l-1,p(i)]} \tag{7}$$
$$P_-^{[l,i]} = \Omega^{[l,i]}\Omega^{[l-1,p(i)]}. \tag{8}$$

Inserting this splitting into (4) leads to a set of projected Schrödinger equations for the tensors $\Lambda^{[l,i]}$ and matrices $R^{[l,i]}$. For example under the action of $P_+^{[l,i]}$ (see also Eq. (5)):

$$i\partial_t\Psi[\alpha] = i\dot{\Lambda}^{[l,i]}(V^{[l-1,p(i)]}V^{[l+1,c_1(i)]}V^{[l+1,c_2(i)]}) + i\Lambda^{[l,i]}\partial_t(V^{[l-1,p(i)]}V^{[l+1,c_1(i)]}V^{[l+1,c_2(i)]})$$
$$= (V^{[l-1,p(i)]}V^{[l+1,c_1(i)]}V^{[l+1,c_2(i)]})H_{eff}^{[l,i]}\Lambda^{[l,i]}, \tag{9}$$

with the effective Hamiltonian environment

$$H_{eff}^{[l,i]} = (V^{[l-1,p(i)]}V^{[l+1,c_1(i)]}V^{[l+1,c_2(i)]})^*\hat{H}(V^{[l-1,p(i)]}V^{[l+1,c_1(i)]}V^{[l+1,c_2(i)]}). \tag{10}$$

We choose a convenient gauge in which the time-derivative of any tensor of the TTNS representation is orthogonal to itself. This must be done to avoid over-completeness of the basis of the tangent space. In this gauge the time derivative simplifies to

$$i\dot{\Lambda}^{[l,i]} = H_{eff}^{[l,i]}\Lambda^{[l,i]}, \tag{11}$$

which is obtained by contracting Eq. (9) with $(V^{[l-1,p(i)]}V^{[l+1,c_1(i)]}V^{[l+1,c_2(i)]})^*$. Similarly, we obtain for $R^{[l,i]}$, which results from the action of $P_-^{[l,i]}$,

$$i\dot{R}^{[l,i]} = \tilde{H}_{eff}^{[l,i]}R^{[l,i]}, \tag{12}$$

with the effective Hamiltonian environment

$$\tilde{H}_{eff}^{[l,i]} = (V^{[l-1,p(i)]}V^{[l,i]})^*\hat{H}(V^{[l-1,p(i)]}V^{[l,i]}). \tag{13}$$

Time-evolution is obtained by integrating the linear differential equations Eqs. (11) and (12) using the projector splitting integrator. Evaluating the action of the Hamiltonian environments in Eqs. (10) and (13) generally requires a compressed representation of the Hamiltonian, for example as a matrix product operator (MPO) or tree tensor network operator (TTNO), in which case the environments are recursively contractible with the TTN. Alternatively one can express the Hamiltonian as a sum of rank-1 terms, in which case evaluating Eqs. (11) and (12) simplifies to a sum over matrix multiplications applied to the tensor for which the time-derivative is calculated. The number of Hamiltonian terms to be evaluated for a given site can be reduced by combining terms in the rank-1 decomposition of the Hamiltonian during the recursive contraction.

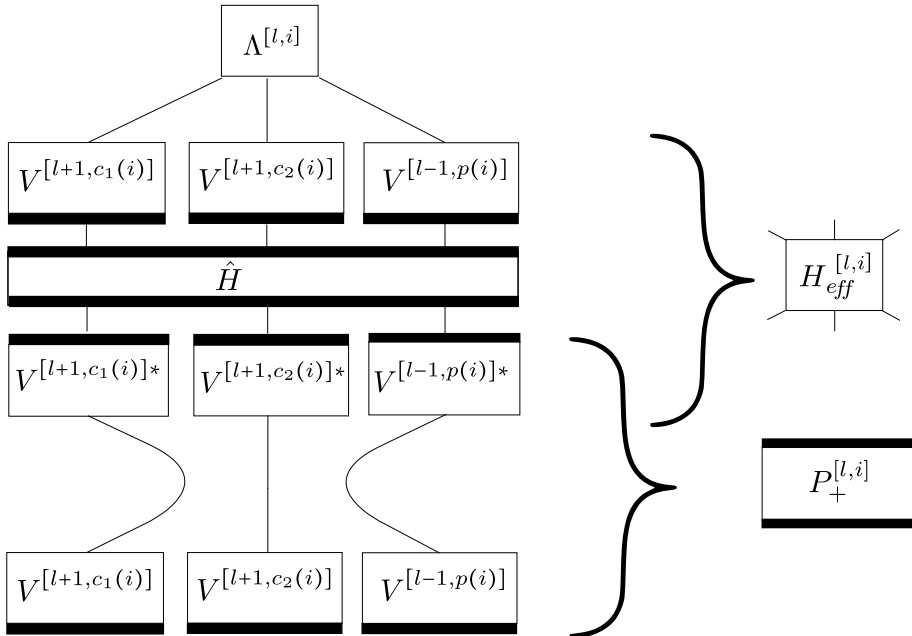

Figure 5: Graphical representation of the last line of Eq. (9), with identification of the effective Hamiltonian environment, $H_{eff}^{[l,i]}$, of Eq. (5) as well as part of the tangent space projector, $P_+^{[l,i]}$, of Eq. (7). In contrast to Fig. 4, the environment tensors have been brought on the same level regardless of layer for better readability.

## 2.3 Splitting integrator

Formally, the splitting integrator is obtained using a Trotter splitting applied to the additive decomposition of the tangent-space-projected evolution operator. Practically, it consists of a forward walk on the tree, propagation of the top-level tensor $\Lambda^{[0,1]}$ for a full time step, and a backward walk on the tree. A pseudo-code is given in algorithms 1-3. During the walks on the tree, isometrization of the TTNS is always maintained about the currently visited node and the effective Hamiltonian matrices are updated when going from one node to another along the direction of the step. The forward walk (backward walk) starts from the top-level node and proceeds from the current node to the adjacent node in a clockwise direction (in a counter-clockwise direction) closest to the previous/incoming node and propagation for half a time step is performed only while ascending (descending). A walk on the tree is finished once the top-node is reached after visiting all physical sites, i.e. after each tensor (and the associated matrix $R$) is propagated save those of the top node.

## 2.4 Remarks

The algorithm introduced above is a generalization of a previously published projector-splitting integrator for TTNS with a single-layer [59,67]. Ref. [63] describes an algorithm for a general TTNS, which is identical to the above algorithm with a single (either forward or backward) walk per time-step. The main differences between the algorithm of Ref. [62] and the one presented here are in the definition of the walk on the tree and in the absence of a top-node, including it's separate propagation routine.

While the TDVP applied to MPS has been demonstrated to be capable of simulating dynamics in two-dimensional systems [21], a detailed analysis and comparison with other tensor network structures is absent in the literature. In particular, the numerical stability of the TDVP cannot be taken for granted [68], especially when interactions between sites are long-ranged and not

---

**Algorithm 1** Forward walk

---

**Input:** $\Psi[l,i]$, current node $[l,i]$, next node $[l-1,p(i)]$
**Output:** $\Psi[l-1,p(i)]$
 1: **if** in forward loop **then**:
 2:     $\Lambda^{[l,i]}(t_{1/2}) \leftarrow \text{propagate}(\Lambda^{[l,i]}(t_0), h/2)$
 3:     compute QR fact. $\Lambda^{[l,i]}(t_{1/2}) = Q^{[l,i]}(t_{1/2})R^{[l,i]}(t_{1/2})$
 4:     $\Lambda^{[l,i]}(t_{1/2}) \leftarrow Q^{[l,i]}(t_{1/2})$
 5:     $R^{[l,i]}(t_0) \leftarrow \text{propagate}(R^{[l,i]}(t_{1/2}), -h/2)$
 6:     $\Lambda^{[l-1,p(i)]}(t_0) \leftarrow \leftarrow Q^{[l-1,p(i)]}(t_0)R^{[l,i]}(t_0)$
 7: **else**
 8:     compute QR fact. $\Lambda^{[l,i]}(t_1) = Q^{[l,i]}(t_1)R^{[l,i]}(t_1)$
 9:     $\Lambda^{[l,i]}(t_1) \leftarrow Q^{[l,i]}(t_1)$
10:     $\Lambda^{[l-1,p(i)]}(t_1) \leftarrow Q^{[l-1,p(i)]}(t_1)R^{[l,i]}(t_1)$
11: **end if**

---

**Algorithm 2** Backward walk

---

**Input:** $\Psi[l,i]$, current node $[l,i]$, next node $[l+1,c_j(i)]$
**Output:** $\Psi[l+1,c_j(i)]$
 1: **if** in backward loop **then**:
 2:     compute QR fact. $\Lambda^{[l,i]}(t_1) = Q^{[l,i]}(t_1)R^{[l,i]}(t_1)$
 3:     $\Lambda^{[l,i]}(t_1) \leftarrow Q^{[l,i]}(t_1)$
 4:     $R^{[l,i]}(t_{1/2}) \leftarrow \text{propagate}(R^{[l,i]}(t_1), -h/2)$
 5:     $\Lambda^{[l+1,c_j(i)]}(t_{1/2}) \leftarrow Q^{[l+1,c_j(i)]}(t_{1/2})R^{[l,i]}(t_{1/2})$
 6:     $\Lambda^{[l+1,c_j(i)]}(t_1) \leftarrow \text{propagate}(\Lambda^{[l+1,c_j(i)]}(t_{1/2}), h/2)$
 7: **else**
 8:     compute QR fact. $\Lambda^{[l,i]}(t_0) = Q^{[l,i]}(t_0)R^{[l,i]}(t_0)$
 9:     $\Lambda^{[l,i]}(t_0) \leftarrow Q^{[l,i]}(t_0)$
10:     $\Lambda^{[l+1,c_j(i)]}(t_0) \leftarrow Q^{[l+1,c_j(i)]}(t_0)R^{[l,i]}(t_0)$
11: **end if**

---

**Algorithm 3** Propagation of top-node's tensor

---

**Input:** $\Psi[0,1](t_0)$
**Output:** $\Psi[0,1](t_1)$
 1: $\Lambda^{[0,1]}(t_1) \leftarrow \text{propagate}(\Lambda^{[0,1]}(t_0), h)$

---

smoothly decaying, as discussed in the following.

The application of TDVP formally requires the TTNS corresponding to the initial condition to possess a full tree rank of **r**. However, many physical initial conditions of interest can be represented with a low rank TTNS or even as a product state. If the initial condition is not contained in the manifold of TTNS with tree rank of **r** due to rank deficiency, the TDVP doesn't provide a prescription for how to choose and evolve the redundant parameters, which will gain weight in the wavefunction representation at later times. Stability and exactness of the dynamics under such circumstances is then dependent on details of the implementation and the model. For the projector splitting integrator, the initial rank-deficiency translates into non-uniqueness of the matrix decompositions employed in the change of isometrization. While the algorithm is not guaranteed to be exact in this case, numerical experiments and prior applications of the algorithm in one-dimensional systems indicate that it is generally reliable even for product state initial conditions. As a check, one may choose to regularize the initial condition by the addition of weak noise, and test for invariance of the resulting dynamics at short times. The initial evolution of redundant variational parameters depends on arbitrary choices such as their initialization, the choice of regularization (if applied), as well as the details of the linear algebra routines used. Thus, different initializations of the same physical state may not converge to the same solution [69,70]. Several approaches have been developed to address this problem. In one-dimensional systems with nearest-neighbour interactions, the commonly used two-site version of the TDVP algorithm of Ref. [49] is free of this issue, although this comes at the cost of breaking unitarity of the evolution when the results cease to be close to the exact solution. For generic interactions and arbitrary TTNS, a scheme to optimally initialize redundant parameters was introduced [70]. However, this scheme requires the evaluation of an effective Hamiltonian matrices for $\hat{H}^2$ and its compatibility with the integration scheme employed here is an open question. Recently, another approach based on a global basis expansion for MPS has been presented, and should also be applicable to general TTNS [68].

Practically, we observe that the dependence of our results on non-optimal initializations of redundant parameters systematically decreases with increasing bond-dimension, which is also expected from the derivation of the optimization scheme mentioned above. The dependence on initialization becomes noticeable only when the wavefunction markedly departs from the exact result, which provides an additional handle to access the convergence of the method.

## 3 Results

We first benchmark the method developed in this work by comparison with exact results obtained for non-interacting fermions on a 2D lattice. In the second stage we propagate a 2D system of hard-core bosons with nearest neighbor interactions and compare our results to propagation using MPS [20]. The mapping of physical sites to the respective tensor network structure is illustrated in Figs. 1 and 2. All calculations employ a regularization of the initial product state, which consists of addition of white noise sampled uniformly from the interval $[-10^{-20}, 10^{-20}]$ and subsequent renormalization of the TTNS.

### 3.1 Non-interacting fermions

We compute the dynamics of non-interacting fermions on a 2D lattice with on-site disorder

$$\hat{H} = J \sum_{<i,j>} \left( \hat{c}_i^\dagger \hat{c}_j + \hat{c}_j^\dagger \hat{c}_i \right) + \sum_i h_i \left( \hat{c}_i^\dagger \hat{c}_i - \frac{1}{2} \right), \tag{14}$$

where the index $i = (x, y)$ indicates the position of the fermion on the lattice, $\langle . \rangle$ is a sum over nearest-neighbours, $h_i$ is drawn from a uniform distribution $[-W, W]$ and $J = 1$. All simulations use an identical initial state which is a random product state at half-filling, and use a time step $dt = 0.01$. The tensor network state calculations employ the Jordan-Wigner transformation of (14)

$$\hat{H} = \sum_{<j,k>,j<k} \hat{S}_j^+ \left( \prod_{j \leq l < k} 2\hat{S}_l^z \right) \hat{S}_k^- + \hat{S}_j^- \left( \prod_{j < l \leq k} 2\hat{S}_l^z \right) \hat{S}_k^+ + W \sum_i h_i \hat{S}_i^z. \tag{15}$$

Different paths along which the sites are enumerated can be chosen, and this choice potentially influences the performance of the TNS algorithm. Here, we choose the path such that the Jordan-Wigner strings span a minimal distance on the graph of the tree tensor network structure. While solving the non-interacting problem in the fermionic representation is trivial, the presence of Jordan-Wigner strings renders its solution with tensor network states just as difficult as that of an interacting problem. We compute the dynamics of this mode both for a clean system ($W = 0$) and for one realization of a moderately strong quenched disorder ($W = 10$). Two-dimensional non-interacting fermions show Anderson localization at any finite disorder strength. While the localization length may exceed the lattice dimensions chosen, disorder nonetheless slows the growth of entanglement and should allow access to longer timescales. Indeed, we observe good agreement for the density profiles, $\hat{n}_{x,y} = \hat{c}_{x,y}^\dagger \hat{c}_{x,y}$ with $x, y \in [1, L]$, along a horizontal cut of the lattice between the exact result and data from both binary and quaternary TNS only up to times $t \leq 1$ for the clean system, while longer times are accessible in the disordered case (see Fig. 6). If the time-step is chosen sufficiently small, errors associated to the linearization of Eq. (4) are negligible compared to inaccuracies related to the finite bond dimensions at all but the earliest times (see lower panel of Fig. 6). To get a more complete picture of the growth of errors with time as well as their dependence on TNS structure and bond dimension, in Fig. 7 we show the average error in the expectation value of the local density as a function of time. Both TNS structures show systematic improvement with increasing bond dimension, and the error grows more mildly at intermediate times in the disordered case. In both cases, smaller deviations from exact results are achievable for binary TTNS than for quaternary TTNS at the employed bond dimensions. We find that a convergence criterion of an average error in the local density of about 2 % agrees well with the qualitative analysis of Fig. 6 and gives a good estimate of the times up to which the TNS results are reliable.

## 3.2 Hard-Core Bosons (XXZ model in 2D)

We consider the dynamics of hard-core bosons on a 2D lattice ,

$$\hat{H} = -J \sum_{<i,j>} \left( \hat{b}_i^\dagger \hat{b}_j + \hat{b}_j^\dagger \hat{b}_i \right) + V \sum_{<i,j>} \hat{b}_i^\dagger \hat{b}_i \hat{b}_j^\dagger \hat{b}_j,$$

with nearest-neighbor interactions and we set $V = J = 1$. We choose an initial condition with a central square sublattice occupied and all other lattice sites empty. This system and initial condition have been studied previously in Ref. [20] using MPS, where results up to $tJ = 2.0$ were presented for a square lattice of linear length $L = 14$. To establish the numerical exactness of the algorithm for this non-integrable model, we compare the results for the local bosonic density $\hat{n}_{x,y} = \hat{b}_{x,y}^\dagger \hat{b}_{x,y}$ with $x, y \in [1, L]$, for both binary and quaternary tensor networks with exact diagonalization for a square lattice of linear length $L = 4$, with the central 4 lattice sites occupied (see Fig. 9). Deviations from the exact result become noticeable only for times $t \approx 2$.

Having established the validity of the algorithm, we investigate the dynamics of an initial product state of a filled, central 4x4 sublattice in a square lattice of a linear length $L = 16$,

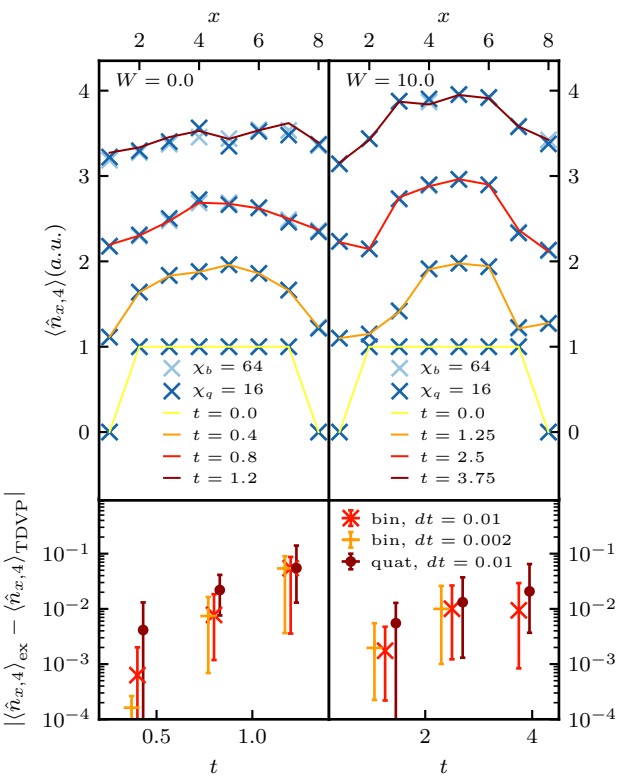

Figure 6: Density profiles of a central, horizontal cut in the fourth row for a random product state configuration of non-interacting fermions on a 8x8 lattice. *Upper panels*: Profiles for $W = 0$ (left) and $W = 10$ (right). Later times are spaced upwards by 1 for readability. TDVP results for binary tensor network with $\chi_b = 64$ (light blue crosses) and quaternary tensor network with $\chi_b = 16$ (dark blue crosses), both with $dt = 0.01$, shown on top of exact results (solid lines). *Lower panels*: The caps of the error bars represent the maximal and minimal deviation of the profiles in the above panels from the exact result for different bond-dimensions, tensor network structures and time-steps.

see Fig. 8. In the upper panel of Fig. 10, we focus on the bosonic density for the site in the fourth row and fourth column of the lattice. In contrast to the non-interacting model and the small two-dimensional lattice discussed above, no exact results are available for this interacting system and $L = 16$. Therefore, the convergence of the results is assessed by comparing the deviation of the local density between different bond dimensions. All examined bond dimensions agree well up to times $t \sim 1.0$. For later times we see agreement for all but the lowest bond dimensions in both quaternary and binary TNS. However, *quantitative* agreement (within a deviation of 0.001) up to $t = 1.5$ only holds between the binary TNS results with $\chi_b = 128$, $\chi_b = 64$ and the MPS results of Ref. [20] at $\chi = 400$ and $\chi = 500$. Since the accuracy of n-ary TTNS can show site-dependence [60], we also report the average density deviation with respect to the best available calculation in the respective TNS structures in Fig. 10. The averaged density supports the observations made for a diagonal site both quantitatively and qualitatively. Particularly, an average deviation of 0.001 is reached at $t = 1.5$ for binary TNS, while quaternary TNS saturate the threshold at $t = 1.2$. The MPS results of Ref. [20] are converged to within this accuracy up to $t = 1.3$, while the deviation between the reference results of both binary TNS and MPS reach the threshold at $t = 1.4$.

Furthermore, since the Hamiltonian and the initial condition are isotropic, distance from

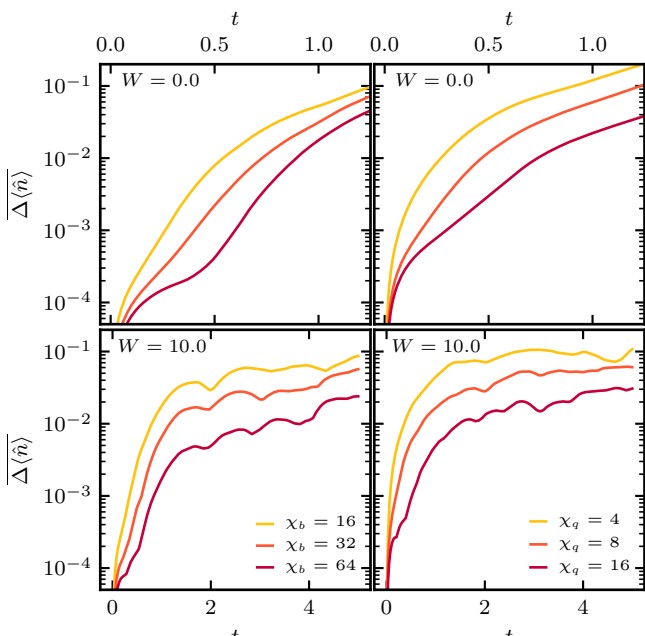

Figure 7: Average deviation from exact $\langle \hat{n}(t) \rangle$ expectation value per site for non-interacting fermions on a clean (top panels) and a disordered (bottom panels, $W = 10$) 8x8 lattice with open boundary conditions. Left panels are binary TTNS and right panels are quaternary TTNS. The time step used is $dt = 0.01$.

the exact solution can also be assessed by the anisotropy $A(t) = \frac{1}{\sum_{x,y=1}^{L} n_{x,y}(t)} \sum_{x,y=1}^{L} \left| \hat{n}_{x,y}(t) - \hat{n}_{y,x}(t) \right|$ of the bosonic density, also reported in Fig. 10. We note however, that while the isotropy of the numerical solution is required, it is not a sufficient condition for the solution to be numerically exact. For both quaternary and binary TTNS, small anisotropies (< 0.3%) are obtained up to their respective convergence times. In Ref. [20], an anisotropy of 4% was reported at $t = 2.0$ using MPS, a threshold which neither binary nor quaternary TTNS saturate at the longest simulated times. Generally, the quaternary TTNS has less anisotropic error since the partitioning of the lattice through the tree structure is isotropic, although the result is less tightly converged than the binary TTNS. Thus, anisotropy is only a useful indicator of convergence when comparing TTNS of the same structure. Given the small deviations in both anisotropy and local densities, we consider our results to be numerically exact up $t = 1.5$ for binary TTNS with $\chi_b = 128$, and up to $t = 1.2$ for quaternary TTNS with $\chi_q = 16$. The performance of the TDVP applied to binary TTNS is thus comparable with the results of Ref. [20], providing the gain of better isotropy of the solution. Note that the bond dimension used for MPS calculations do not correspond to the current state-of-the-art, and larger bond dimension may be feasible for binary TNS when using symmetries of the Hamiltonian. Due to the lack of an exact solution to compare to, the convergence criterion employed is significantly tighter than in the case of free fermions to ensure quantitatively accurate results.

# 4   Discussion

In this work we have assessed the performance of TTNS for simulating the dynamics of two-dimensional many-body lattice systems. We introduced an algorithm based on the time-dependent variational principle for arbitrary TTNS and benchmarked it on systems of non-

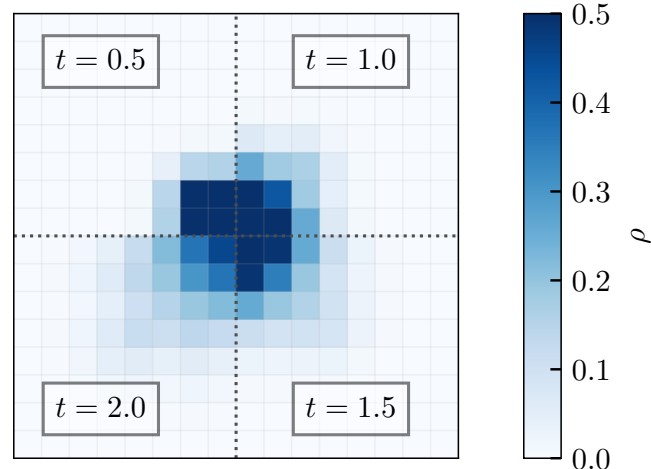

Figure 8: Spreading of hard-core boson density $\langle \hat{n}_{x,y} \rangle$, initially occupying the central 4-by-4 sublattice of a square lattice with $L = 16$. Time step used is $dt = 0.01$, and the scale is restricted to a maximum of $n_i = 0.5$ for clarity.

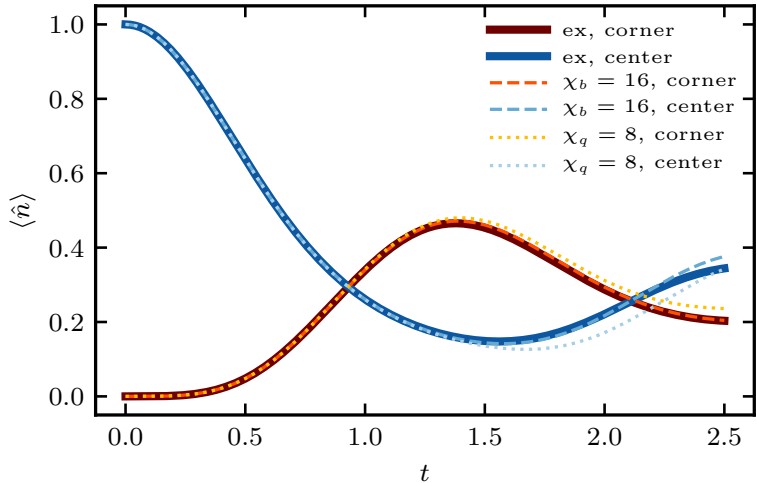

Figure 9: Bosonic site density as a function of time for a 4x4 lattice with the central 2x2 sites filled at $t = 0$. Two special sites are shown (corner and central). Exact results (solid lines) and TNS results for binary (dashed lines) and quaternary (dotted lines) TNS. Time step used for both panels is $dt = 0.01$.

interacting fermions and interacting hard-core bosons in two dimensions, comparing the performance to previously published results using matrix product states. During the preparation of the manuscript we became aware of a recent complementary work introducing a similar versions of the algorithm, which were applied in rather different settings (as an impurity solver [62], and in a more formal derivation of the algorithm [63]).

Currently, no efficient technique exists for exactly simulating the non-equilibrium dynamics of interacting, two-dimensional quantum systems. Despite recent progress, the timescales accessible by tensor network techniques for such systems are generally extremely short. We have found tree tensor networks to perform at least as well as matrix product state techniques, with binary TTNS generally providing a more robust performance than their quaternary

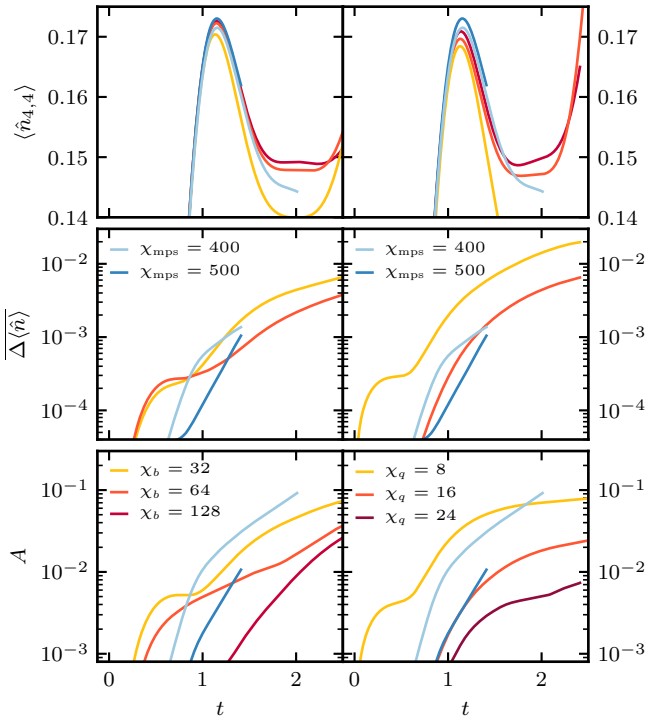

Figure 10: Measures of convergence for hard-core bosons in 16x16 lattice for binary (left panels) and quaternary (right panels) TTNS as well as MPS [20] (all panels, blue shades). *Upper panels*: Bosonic density for the 4th left and 4th topmost site. *Middle panels*: Average deviation of the local bosonic density with respect to best available result within the respective TNS structure, for binary TTNS and MPS (left panel) as well as quaternary TTNS and MPS (right panel). For $\chi_{\mathrm{mps}} = 500$, the deviation is reported with respect to $\chi_b = 128$. . *Bottom panels*: Anisotropy (see text) of bosonic density. The time step used is $dt = 0.01$.

counterparts. The issue of analyzing the convergence, and thus ensuring the numerical exactness of the computed result, was discussed. We believe the availability of an alternative to matrix product states in the form of more general TTNS is important and can offer additional insight in situations when slow convergence is observed.

Our analysis has been mostly qualitative and a promising future avenue is the exploration of the entanglement structure of out-of-equilibrium states in 2D lattices . This will aid in the identification of optimal tensor network structures in order to best exploit the increased flexibility of TTNS, which already has proven to be important in applications for zero-dimensional systems, such as impurity models and also for molecular quantum dynamics [61, 62, 71–73]. The dynamics of one-dimensional systems quenched to a critical point is another application where such an increased flexibility may be of advantage. For critical systems in equilibrium, the multi-scale entanglement renormalization (MERA) [74,75] ansatz provides an efficient tensor network structure, which bears resemblance with the *n*-ary tree structures employed here. However, since a time-evolution approach for MERA is missing, it is interesting to compare the performance of MPS and *n*-ary TTNS for critical systems out-of-equilibrium. We leave such an investigation to a future work.

## Acknowledgments

We thank Frank Pollmann for providing raw data from Ref. [20]. BK and DRR acknowledge funding through the National Science Foundation Grant No. CHE-1954791. YBL acknowledges support by the Israel Science Foundation (grants No. 527/19 and 218/19).

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
