# Peer review of "Studying dynamics in two-dimensional quantum lattices using tree tensor network states"

_SciPost Physics, doi:SciPost Phys. 9, 070 (2020)_

## Round 2 · Referee Report · Matteo Rizzi (Referee 1) · 2020-5-8

Strengths

1- Tackling time-evolution of two-dimensional systems is an open problem, where tentatives have been done, but no standard recipe is available yet -- therefore definitely with topic to be further investigated

2- The manuscript is self-contained, with a large didactic introduction of the whole notation (see however weak point 3) and even a pseudo-code snippet

3- The chosen benchmarks are fully reasonable: an exactly solvable problem (non-trivial for tensor-network (TN) methods, though), and another system recently tackled in similar studies

4- The overall structure of the manuscript and the aims of the work are clear, and there is no apparent trace of over-selling results

Weaknesses

1- The analysis of the benchmark results is not clear in some aspects, and seems performed in a bit of a rush -- see Report for Details

2- Results are presented for two different tree-tensor-network (TTN) structures, but the "quaternary" is not mentioned or sketched in the whole text (though it is easy to imagine what is that...)!

3- A couple more illustrations in Section II, where quite some definitions and symbols are introduced, would be desirable: without knowing Ref. 36 almost by heart, it would be quite though for the reader to reconstruct the derivation of the method.

Report

As highlighted in the list of "Strengths", the manuscript deals with an interesting problem, the description of time-evolution in two-dimensional quantum systems via tensor-network methods, for which a solution is far from being fully established.
Moreover, it illustrates the method and the results of sensible benchmarks in a self-contained manuscript.
Therefore it definitely has the potential for being published.
However, as summarised under "Weaknesses", there are a number of reasons that prevent its publication in its present form and demand a further iteration by the Authors.
After having addressed them, I would be in favour of publication.
* * *
In general, the manuscript seems to give proper credit to other contributions and to state similarities and differences with works dealing with a very similar problem (11-21 & 35-41). Only I was a bit surprised not to find some more works about trying to extend the limitations of time-evolution, e.g.,
C. Krumnow, L. Veis, Ö. Legeza, and J. Eisert, Phys. Rev. Lett. 117, 210402 (2016);
C. Krumnow, J. Eisert, and Ö. Legeza, arXiv:1904.11999;
M.M. Rams and M. Zwolak, Phys. Rev. Lett. 124, 137701 (2020)
and under the ones listed about "TTNS are rarely used in the context of interacting lattice systems", e.g.,
V. Murg, F. Verstraete, Ö. Legeza, and R. M. Noack, Phys. Rev. B 82, 205105 (2010)
W. Li, J. von Delft, and T. Xiang, Phys. Rev. B 86, 195137 (2012)
M. Gerster, et al., Phys. Rev. B 96, 195123 (2017)
E. Macaluso, et al., Phys. Rev. Research 2, 013145 (2020)
Yes, the latter are own references, sorry about that, but they deal with 2D strongly correlated systems and thus fit particularly well in such a list. A more extensive search in the literature is definitely welcome.
* * *
Some statements could probably be sharpened: e.g., "Since the entanglement of a generic system grows linearly with time, the accessible timescales are limited" should be phrased in terms of global quenches, and the distinction with respect to local ones made, etc.
By the way, Ref. 1 was among the first ones providing numerical evidence, but there are theoretical predictions of the linear growth in 1D, too:
e.g., P. Calabrese and J. Cardy, J. Stat. Mech.: Theory and Experiment 2005, P04010 (2005) and Journal of Statistical Mechanics: Theory and Experiment 2007, P10004 (2007) (the second one focussing on the mentioned distinction between local and global quenches).
* * *
A graphical representation of Eqs. 5-8 would be desirable for the readers not immediately recalling the whole Ref.36 by heart
Incidentally, a graphically more polished version of Figs. 1-3 could be nice to see (though only decorative, I admit).
* * *
About the time evolution procedure itself:
i) Should Eq. (12) not contain a minus prefactor with respect to Eq. (11), according to Eq.(5) and to what happens for the MPS case in Ref.36?
ii) Below Eq. 12 it is stated that, in general, "the result of the contraction of the Hamiltonian with the environment tensors is not a set of matrices but a compact tensor network". Do the Authors mean, as it should be the case, that a MPO/TPO Hamiltonian would result in a MPO/TPO connection of the three terms in Fig.3b? Why then not illustrate it in Fig. 3 directly, and simplify considerably the discussion?
iii) I must admit that in Sec.II.C-D I do not see the difference between the adopted "splitting integrator" and "propagation carried out while descending and ascending on the tree": Could the Authors clarify their point, possibly via an illustration?
iv) What supports the statement "it is unlikely that the quadratic error bound in the time-step for the total time-evolution of the latter carries over"? It would be highly desirable to substantiate this with a quantitative analysis in the following, which I cannot find.
v) Same applies for "This intuition, however, is largely based on systems with local, or at least smoothly decaying interactions, and can break down for interactions which result from mapping a 2D lattice to a 1D chain, for example."
vi) I find the remarks about the "redundant parameters" scheme interesting, but how does this procedure compare with a two-tensor optimization, or a local inflation of the bond dimension in a one-tensor scheme, as it is done for optimization purposes already? [see White, Phys. Rev. B 72, 180403 (2005), Hubig, et al., Phys. Rev. B 91, 155115 (2015); Silvi, et al., SciPost Phys. Lect. Notes 8 (2019)]
* * *
Detailed remarks about the analysis of the benchmark:
a) The role of the time-step is not discussed and clearly separated from the approximation due to bond-dimension truncation.
b) The statements about better reliability of the binary structure with respect to the quaternary one do not seem to be supported from the plots in Fig. 6, unless I am overlooking something very naive. In particular, I really do not see why the Authors say that data from the binary structure display a "systematic improvement wrt bond dimension", implying that the ones from quaternary do not...
c) An hint about the relative error, not only its absolute value, would be desirable; even better would be the typical plot of the exact (or best) solution vs all other approximate curves as a function of time: I mean in the style that is typically used to illustrate the explosion of errors in a 1D setup -- this would probably help about point a), too. At this level, and in view of the point iv) above, it is not clear that the choice of integration method and time-step does not play an important role, too...
d) The results for non-interacting fermions are all for an initial random state (by the way, at which average filling?): What would be different for a more structured initial state? Say a cluster away from or around the sample boundaries, a checkerboard pattern, etc.? At least one would naively expect to see a comparison between fermions and hard-core bosons in the same style of Fig.5, not?
e) Maybe a too ambitious question, but would other kind of transformations, like Bravyi-Kitaev instead of Jordan-Wigner, impact on the performances of the algorithm?
f) Incidentally, are symmetries, namely particle-number conservation, encoded in the TTNs structure?
g) Why do the Authors not ask the raw data of Ref.16 for a direct comparison? This would be of great interest, as stated somewhere in the text, too.
* * *
Minor points:
- At line 5 of Algorithm 2, it should read t_{1/2} instead of t_0, right?
- What does "convergence of the local density within 2" mean at page 6, column 1?
- A part of the caption of Fig.4 got probably lost: it should read similar to Fig.6, in which the reference to quaternary TTNs is clear (this should be added in the main text, too).
- When comparing the best binary and quaternary data, the Authors seem to infer a lot from their difference being of order 1%: where exactly is the surprise, given that both have an error estimate of roughly that order? Can the Authors find a better illustration to convey their message?
- Alongside with refs. 36 and 41 on the TDVP method, I would have expected to see also J. Haegeman, et al., Phys. Rev. Lett. 107, 70601 (2011): is there a particular reason behind the choice of omitting it?
- I am honestly puzzled by "For times beyond the convergence time we also observe a strong influence ...": are data of any meaning beyond that point? Maybe it is a mismatch in what I understood, but the term "convergence time" is not defined anywhere...
- What do the Authors exactly intend with dynamics of "critical one-dim. systems"? What is going to vary and what is quenched or tuned?
- By the way, there are good reasons for a stable time-evolution algorithm be missing for MERAs (though notice Phys. Rev. A 77, 052328 (2008)), given that their structure is full of loops and already extremely hard to manipulate for static optimisations...

---

## Round 2 · Referee Report · Luca Tagliacozzo (Referee 2) · 2020-5-21

Strengths

The manuscripts propose to use tree tensor networks to perform the time evolution of a 2D quantum system. There could be several good reasons to follow such a procedure but very few of them are carefully reported.

Weaknesses

The main weakness is
1) the lack of a critical analysis of their results,
2) the lack of an accurate benchmark of their algorithm (against i.e. ground-state physics)
3) the consideration of only single-site operators
4) the lack of description of the physics of the models considered (e.g. location of the localization transition etc)

Report

I have read the paper with interest and cannot recommend it for publication in its present form.
The authors implement the TDVP for tree tensor network (something that was already done in the literature) and use it to time evolve 2D states.

They constantly mention exact results, but their benchmarks are incomplete, show a strong dependence on the bond dimension of the TTN and there is no criterium the reader can use to establish when the results are reliable.

As a side remark they miss several key references on the topic (including those where the TDVP has been used to optimize 2D TTN), and seem to be completely unaware of the current trends in the TN community where people try to understand how to establish when the results of approximate time evolution are reliable. On this specific aspect, I suggest they read the introduction and references in our recent paper J. Surace, M. Piani, and L. Tagliacozzo Phys. Rev. B 99, 235115 – Published 7 June 2019 where we have tried to give an overview of the effort in the community.

In the following I list the concerns that lead me to formulate the above opinion in detail starting from the analysis of the results presented.

**Regarding the results,

The authors start studying an exactly solvable case of free fermions with disorder.
I am quite puzzled that there is no plot of the exact evolution but only of the error with respect to the exact result.

This does not provide a good insight on how large or small the error is (since it is not even the relative error they plot).
1) They should add the plot of the exact results (even if boring) and superimpose the results of their simulations on the top of it, at least in a inset of their current plots.

2) What convergence within two means in the following sentence?
" Without disorder, convergence of the local density within 2 is obtained for both quaternary and binary TTNS only for t ≤ 1.
They decide to only characterize local expectation values of the fermionic occupation (something pretty restrictive given that they have access to the full many-body wave function). Anyway there is a complete lack of discussion about what are the expected results for such
local observable. I would assume that at least for weak disorder they should equilibrate to a value given by the Gaussian Diagonal Ensemble constructed from their initial state.

3) What is the equilibration time?

4) Is it shorter than the recurrences?

5) When the disorder is ramped up, they enter an Anderson localized phase, what is the critical strength for the transition?

6) What is the effect of such a transition in their evolution? Could they observe it?

7) The values of the used bond dimensions are far too small (in the binary tree they can be pushed to several hundred/ a thousand), they should repeat the simulations with larger bond dimension until lines for the errors with different bond dimension superimpose and then they can attribute all the remaining deviation from the exact evolution to the Trotterization errors.

8) Given the availability of exact results in this context, why haven't the authors tried to extrapolate the finite bond dimension results and see if they are able to obtain the exact ones?

9) I am also concerned with the lack of fundamental checks on their evolution. For example the evolution is unitary and should conserve the energy of the initial state. Actually the deviation of the conservation of the energy has been widely used as a measure on how good an approximate evolution is.
In the present version there are no plots about the energy conservation, and the lack of discussion about the use or not of a symplectic integrator and if and for how long the energy is conserved.
 
10) As a result the plots presented are not really conclusive, they show that if you run the tdvp on tree tensor network you get some results out of it. But what these results mean is far from obvious, at least for me. How does this approach compare with say a simple spin-wave analysis or semiclassical approaches like the truncated Wigner methods?

11) Using the Jordan Wigner transformation for simulating fermions is definitely a possibility but I would suggest at least to mention all the theoretical development about fermionic tensor networks.

12) The sentence  ."..The application of TDVP formally requires the TTNS corresponding to the initial condition to possess a full tree rank of r." is formally wrong. The fact that TDVP works on subspaces with fixed bond dimension has been addressed already in the literature by passing to a two sites algorithm (and some initialization stage). I believe that adding noise to the initial state in order to fill the rank is a pretty dangerous strategy, that could lead to unexpected results
My belive seems to be confirmed by the sentence
"For times beyond the convergence time we also observe a strong influence on the initialization procedure
of the redundant parameters of the TTNS. Increasing the bond dimension systematically reduces this effect."

 13) What is the convergence time they talk about?
All the above observations are even more relevant once they study the non-solvable system of hard-core bosons. Here we are completely lost and do not know how to address the results.

14) The quantity rho discussed in the text is defined in a label of fig. 5 with a site index (already pretty strange, I would have expected a definition in the main text). Furthermore in the definition of the anisotropy A \rho is defined with two indices that are summed over (so they supposedly represent the two coordinates of a point). However it is not clear to me where the zero of coordinates is and thus which asymmetry A is measuring.
I guess it is the crossing of the two lines in fig 5 but the reader should not guess.

15) Furthermore there is no explicit indication about how physical sites in the lattice are mapped to the TTN, since one could do it in infinitely many ways. I guess that the obvious choice has been made, but it would be important to understand if the 0,0 point is in between the tensors or in the center of a 4 leg tensor, in order to understand better the role of the structure of the network on this geometry.

16) I am very puzzled by the sentence "However, given the systematic improvement in the convergence with respect to the bond dimension combined with reduction of the anisotropy of
the results, we consider the latter to achieve numerically exact results for t < 2." 
Their plots show huge variations between TTN with different bond dimensions and between TTN with different structures, what is exact from their point of view?

***General mistakes and lack of references
First of all there are several errors in the text.

1) In the introduction they claim "the logarithm of the latter quantity gives an upper bound on the entanglement entropy for every bipartition of the lattice" this is simply wrong and should be reformulated correctly. (Think of a bipartition in an MPS when one separates even from odd sites)

2) Saying that PEPS algorithms involve uncontrolled approximations seems to me a bit too strong. But this is actually an opinion.

3) There is a typo "however describing to higher spatial dimension"

4) In the list of reference about MPS with long-range interactions key references are missing, in particular, the foundational papers are as far as I am aware

Gregory M. Crosswhite, A. C. Doherty, and Guifré Vidal
Phys. Rev. B 78, 035116 – Published 14 July 2008

Fröwis, F.;Nebendahl, V.;Dür, W.
Physical Review A, vol. 81, Issue 6, id. 062337

5) I have co-authored two papers in this context that deal with 1) the TDVP for systems with long-range interaction, in a version where the MPS is interpreted as a TTN (as you can appreciate in the drawings in the supplementary material). The algorithm is characterized both at equilibrium in  Koffel, Lewenstein, Tagliacozzo, Physical Review Letters, vol. 109, Issue 26, id. 267203
and out of equilibrium in  Hauke Tagliacozzo  Physical Review Letters, vol. 111, Issue 20, id. 207202.
In the supplementary material of both papers we have introduced a version of the single-site TDVP these two papers should be cited in conjunction with References 16 and 36 since they both were published before the two cited papers.

6) The TDVP for studying ground states of 2D TTN was implemented and benchmarked in Andrew J. Ferris Phys. Rev. B 87, 125139 – Published 25 March 2013  the paper should also be referenced.

7) As a matter of fact, I strongly recommend using the TDVP they implement to first find the ground state of 2D systems described with TTNs (by just moving to imaginary time) and compare the results they obtain with the one available in the literature about bond-dimensions and precision (e.g. the above paper is a good starting point but they could also compare with the results of their reference 23).

Once they are sure that their algorithm does not contain any error, they can use it to time evolve the systems and perform accurate comparisons with free systems (as indicated in the first part of this report) and then present the results for the interacting cases.

Alternatively, if they do not want to compare their algorithm with known results at equilibrium, they should at least compare it carefully with the exact results, where available, and run extensive tests against exact diagonalization results on small lattices where the analytical results are not available.

Requested changes

They are already listed in detail in the previous

---

## Round 2 · Referee Report · Anonymous (Referee 3) · 2020-6-1

Strengths

1- The manuscript is overall well written and and the results are critically discussed,

2- Finding efficient methods to simulate the real time evolution of D>1 dimensional systems is a timely and important challenge.

Weaknesses

1- Lack of references (many of them have already been mentioned by the other two referees). Also the isometric form had been used early on D. Nagaj, E. Farhi, J. Goldstone, P. Shor, and I. Sylvester, Phys. Rev. B 77, 214431 (2008).

2- From the introduction it is not getting completely clear why we should expect tree-tensor networks to perform better than matrix-product states. In particular, they have the same 1D area law restrictions.

3- Given that the manuscript promises to provide a benchmark, a more quantitative comparison would be useful.

Report

In my opinion, the work represents a useful exploration of tree-tensor networks as a tool to study the real time dynamics. While I found the results not too surprising, it is still useful to have a benchmark of the method.

Requested changes

The existing reports already include all my criticisms, and thus I have no additional requests.

---

## Round 3 · Referee Report · Matteo Rizzi (Referee 1) · 2020-8-7

Report

A large part of the issues raised by me and the other Referees have been addressed in a satisfactory manner in this revised version, in particular a clearers description of some aspects of the method and a more detailed comparison of numerical results with other methods.
For the remaining few a sensible motivation not to tackle them in the present work has been provided, in my honest opinion, and will remain available for the readership anyway.

Overall, I think that the paper is publishable in its present form.
Certainly, it could have been good to push a bit more in certain directions asked by Ref. 2, but it is his decision to assess how satisfactory were the answers to him, and to possibly ask for a deeper revision.

---

## Round 3 · Referee Report · Luca Tagliacozzo (Referee 2) · 2020-10-5

Report

I find that the answersto the previous reports are adequate. Together with them, the authors improved substantially their initial work and added relevant references.
Furthermore, when they did not modify the manuscript, the existence of a discussion about the points raised will help the reader to assess the relevance of the results and put the paper in context. I thus am pleased of the process and am sure that the manuscript is now suited for publication is Scipost.

Regarding ScipostPhysics, I am also asked to take into account the Acceptance criteria, since it is the flagship Journal in Scipost.
As expressed in the discussion, I do not feel that the manuscript is groundbreaking.

It however presents some benchmark for using tree-tensor networks out of equilibrium, something that should have been done, and it is done here.
For these reasons, I find it very interesting.

I thus leave to the Editor, who is an expert in the field, the choice about how much the paper fulfills point 2-
Present a breakthrough on a previously-identified and long-standing research stumbling block- of the acceptance criteria for Scipost Physics.

One can make a case it does since it presents an out-of-equilibrium simulation of a 2D system, however, we already know that the ansatz used has the same limitations and pitfalls of the MPS and thus its suitability for large times and systems is still unclear.

---

## Round 3 · Referee Report · Anonymous (Referee 3) · 2020-10-14

Report

In my opinion, the criticism raised by my and the other reports have been addressed and I recommend publication of the revised manuscript.

---

## Round 3 · Author Response

Responses to Referee 1

Strengths 1- Tackling time-evolution of two-dimensional systems is an open problem, where tentatives have been done, but no standard recipe is available yet -- therefore definitely with topic to be further investigated 2- The manuscript is self-contained, with a large didactic introduction of the whole notation (see however weak point 3) and even a pseudo-code snippet 3- The chosen benchmarks are fully reasonable: an exactly solvable problem (non-trivial for tensor-network (TN) methods, though), and another system recently tackled in similar studies 4- The overall structure of the manuscript and the aims of the work are clear, and there is no apparent trace of over-selling results Weaknesses 1- The analysis of the benchmark results is not clear in some aspects, and seems performed in a bit of a rush -- see Report for Details 2- Results are presented for two different tree-tensor-network (TTN) structures, but the "quaternary" is not mentioned or sketched in the whole text (though it is easy to imagine what is that...)! 3- A couple more illustrations in Section II, where quite some definitions and symbols are introduced, would be desirable: without knowing Ref. 36 almost by heart, it would be quite though for the reader to reconstruct the derivation of the method. Report As highlighted in the list of "Strengths", the manuscript deals with an interesting problem, the description of time-evolution in two-dimensional quantum systems via tensor-network methods, for which a solution is far from being fully established. Moreover, it illustrates the method and the results of sensible benchmarks in a self-contained manuscript. Therefore it definitely has the potential for being published. However, as summarised under "Weaknesses", there are a number of reasons that prevent its publication in its present form and demand a further iteration by the Authors. After having addressed them, I would be in favour of publication.

In general, the manuscript seems to give proper credit to other contributions and to state similarities and differences with works dealing with a very similar problem (11-21 & 35-41). Only I was a bit surprised not to find some more works about trying to extend the limitations of time-evolution, e.g., C. Krumnow, L. Veis, Ö. Legeza, and J. Eisert, Phys. Rev. Lett. 117, 210402 (2016); C. Krumnow, J. Eisert, and Ö. Legeza, arXiv:1904.11999; M.M. Rams and M. Zwolak, Phys. Rev. Lett. 124, 137701 (2020) and under the ones listed about "TTNS are rarely used in the context of interacting lattice systems", e.g., V. Murg, F. Verstraete, Ö. Legeza, and R. M. Noack, Phys. Rev. B 82, 205105 (2010) W. Li, J. von Delft, and T. Xiang, Phys. Rev. B 86, 195137 (2012) M. Gerster, et al., Phys. Rev. B 96, 195123 (2017) E. Macaluso, et al., Phys. Rev. Research 2, 013145 (2020) Yes, the latter are own references, sorry about that, but they deal with 2D strongly correlated systems and thus fit particularly well in such a list. A more extensive search in the literature is definitely welcome.

The references were added to the introduction.

Some statements could probably be sharpened: e.g., "Since the entanglement of a generic system grows linearly with time, the accessible timescales are limited" should be phrased in terms of global quenches, and the distinction with respect to local ones made, etc. By the way, Ref. 1 was among the first ones providing numerical evidence, but there are theoretical predictions of the linear growth in 1D, too: e.g., P. Calabrese and J. Cardy, J. Stat. Mech.: Theory and Experiment 2005, P04010 (2005) and Journal of Statistical Mechanics: Theory and Experiment 2007, P10004 (2007) (the second one focussing on the mentioned distinction between local and global quenches).

We made the statement more precise by restricting it to global quenches and added one of the mentioned references. We decided to omit a technical discussion of the difference between the dynamics of the entanglement entropy for global and local quenches in the introduction to keep the introduction focused and concise.

A graphical representation of Eqs. 5-8 would be desirable for the readers not immediately recalling the whole Ref.36 by heart Incidentally, a graphically more polished version of Figs. 1-3 could be nice to see (though only decorative, I admit).

We modified Fig. 3 (Fig. 5 in the revised manuscript) and modified Eqs. 5-8 slightly. Particularly, Fig. 3 includes a complete graphical description of the last line of Eq. 9, from which the time derivative of a local tensor is obtained. It also identifies the tangent space projector involved as well as the shorthand for the effective Hamiltonian environment. Furthermore, we now describe the step required to go from Eq. 9 to Eq. 11, in the text.

About the time evolution procedure itself:i) Should Eq. (12) not contain a minus prefactor with respect to Eq. (11), according to Eq.(5) and to what happens for the MPS case in Ref.36?

Since we switch the limits of integration (R is propagated backwards in time, due to the minus sign in Eq. (5)), the absence of a minus sign in Eq. 12 is consistent. ii) Below Eq. 12 it is stated that, in general, "the result of the contraction of the Hamiltonian with the environment tensors is not a set of matrices but a compact tensor network". Do the Authors mean, as it should be the case, that a MPO/TPO Hamiltonian would result in a MPO/TPO connection of the three terms in Fig.3b? Why then not illustrate it in Fig. 3 directly, and simplify considerably the discussion?

Indeed, we changed the notation for the effective Hamiltonian environment accordingly and streamlined Fig. 3 (in the original manuscript, Fig. 5 in the revised version.). iii) I must admit that in Sec.II.C-D I do not see the difference between the adopted "splitting integrator" and "propagation carried out while descending and ascending on the tree": Could the Authors clarify their point, possibly via an illustration?

The original phrasing, besides being unclear, was incorrect due to a minor misinterpretation of the algorithm of [62]. The difference between the two algorithms is only in the definition of the walk on the tree, and we think it does not warrant an extensive technical discussion in the manuscript. iv) What supports the statement "it is unlikely that the quadratic error bound in the time-step for the total time-evolution of the latter carries over"? It would be highly desirable to substantiate this with a quantitative analysis in the following, which I cannot find.

The support for the statement comes from the rigorous derivation of the algorithm in [63], which proves only a linear error bound (in contrast to the quadratic one for MPS). In particular, [63] constructs the algorithm in terms of nested subflows on subtrees, which are only solved approximately. While this does not contradict the exactness property of the algorithm, it may lead to the adjoint of the algorithm not being exactly evaluable. The latter is a requirement for the symmetric combination of first-order integrator and its adjoint to give a second order integrator, as discussed in the appendix of [49]. Thus the naive approach of reverting the direction of the walk on the tree (in analogy to the symmetric algorithm for MPS), taken both in this work and in [62], is unlikely to yield a proper second order integrator. Unfortunately, investigating the scaling of the error with the timestep is non-trivial due to the discussed issues/technicalities associated with the TDVP, especially at short times. However, [62] provides numerical evidence for a quadratically scaling error with timestep in the limit of large timesteps. We decided to not burden the reader with a technical, but necessarily vague discussion and analysis of timestep errors and removed the discussion from the text. [49] Jutho Haegeman, Christian Lubich, Ivan Oseledets, Bart Vandereycken, and Frank Verstraete Phys. Rev. B 94, 165116 – Published 10 October 2016 [62] Daniel Bauernfeind, Markus Aichhorn, SciPost Phys. 8, 024 (2020) · published 7 February 2020 [63] G. Ceruti, C. Lubich, and H. Walach, arXiv preprint arXiv:2002.11392 (2020)\ v) Same applies for "This intuition, however, is largely based on systems with local, or at least smoothly decaying interactions, and can break down for interactions which result from mapping a 2D lattice to a 1D chain, for example." vi) I find the remarks about the "redundant parameters" scheme interesting, but how does this procedure compare with a two-tensor optimization, or a local inflation of the bond dimension in a one-tensor scheme, as it is done for optimization purposes already? [see White, Phys. Rev. B 72, 180403 (2005), Hubig, et al., Phys. Rev. B 91, 155115 (2015); Silvi, et al., SciPost Phys. Lect. Notes 8 (2019)]

We added a discussion of all such procedures (to the best of our knowledge) in the context of applications of the TDVP, which includes a recent work inspired by the local inflation of the bond dimension (applying a global inflation scheme though) [68]. The latter also provides evidence for the potential failure of the TDVP for non-smooth interactions. In order to limit the technicality of the discussion, we decided to not provide numerical evidence for the latter and removed the remark about non-smoothly decaying interactions. [68] 5 M. Yang and S. R. White, “Time dependent variational principle with ancillary krylov subspace,” (2020), arXiv:2005.06104 [cond-mat.str-el]

Detailed remarks about the analysis of the benchmark: a) The role of the time-step is not discussed and clearly separated from the approximation due to bond-dimension truncation.

We now provide additional results for a smaller timestep for the non-interacting fermion calculations, showing that the error is only relevant in comparison to the error due to finite bond dimension at very early times. Analysing the scaling of the error with timestep at early times is an involved problem, due to the intricacies discussed in the remarks on the algorithms. Since timestep errors are generally easily controlled and checked for, we decided to not include a more detailed analysis of the timestep error. b) The statements about better reliability of the binary structure with respect to the quaternary one do not seem to be supported from the plots in Fig. 6, unless I am overlooking something very naive. In particular, I really do not see why the Authors say that data from the binary structure display a "systematic improvement wrt bond dimension", implying that the ones from quaternary do not... c) An hint about the relative error, not only its absolute value, would be desirable; even better would be the typical plot of the exact (or best) solution vs all other approximate curves as a function of time: I mean in the style that is typically used to illustrate the explosion of errors in a 1D setup -- this would probably help about point a), too. At this level, and in view of the point iv) above, it is not clear that the choice of integration method and time-step does not play an important role, too...

We are thankful for the referee’s constructive criticism which helped us to improve our discussion considerably. We revised the discussion of the convergence of the result, including a tighter convergence criterion, motivated by a comparison of the evolution of the bosonic density for a representative site for different bond dimensions. Both the latter and the deviation averaged over the whole lattice show a tighter convergence for the binary TTNS compared to the quaternary TTNS at the bond dimensions used, which leads us to the conclusion that the binary TTNS data is converged up to later times. d) The results for non-interacting fermions are all for an initial random state (by the way, at which average filling?): What would be different for a more structured initial state? Say a cluster away from or around the sample boundaries, a checkerboard pattern, etc.? At least one would naively expect to see a comparison between fermions and hard-core bosons in the same style of Fig.5, not?

We choose a random initial state (at half filling on average) as it is expected to be an unbiased test for different TTNS structures. For specific initial conditions, it is possible to tailor both the structure of the TTNS and the mapping of physical lattice sites to the leaf nodes of the TTNS in order to represent the entanglement buildup most efficiently. This was, however, not attempted here. e) Maybe a too ambitious question, but would other kind of transformations, like Bravyi-Kitaev instead of Jordan-Wigner, impact on the performances of the algorithm?

This is an interesting proposal beyond the current scope of the manuscript and may be worth a follow-up investigation. f) Incidentally, are symmetries, namely particle-number conservation, encoded in the TTNs structure?

Symmetries are currently not implemented (a design decision/flaw, the implementation of symmetries for TTNS is analogous to implementations for MPS). g) Why do the Authors not ask the raw data of Ref.16 for a direct comparison? This would be of great interest, as stated somewhere in the text, too.

We thank the referee for this suggestion, which simplified our convergence analysis and supports the validity of our results further.

Minor points: - At line 5 of Algorithm 2, it should read t_{1/2} instead of t_0, right?

We thank the referee for pointing out this typo, it has been corrected.

  • What does "convergence of the local density within 2" mean at page 6, column 1?

We missed a ‘%’ here, and corrected this typo.

  • A part of the caption of Fig.4 got probably lost: it should read similar to Fig.6, in which the reference to quaternary TTNs is clear (this should be added in the main text, too).

We corrected the caption.

  • When comparing the best binary and quaternary data, the Authors seem to infer a lot from their difference being of order 1%: where exactly is the surprise, given that both have an error estimate of roughly that order? Can the Authors find a better illustration to convey their message?

Indeed, the conclusions drawn previously from the slightly larger deviation were not justified. While a similar conclusion still holds in our improved analysis, discussed above, the inclusion of absolute data for a representative site now motivates a tighter convergence criterion. In particular, the tighter convergence criterion leads to a stronger separation of the error estimates for different TNS structures at the convergence time.

  • Alongside with refs. 36 and 41 on the TDVP method, I would have expected to see also J. Haegeman, et al., Phys. Rev. Lett. 107, 70601 (2011): is there a particular reason behind the choice of omitting it?

We included the reference in the revised version. It was originally omitted because it discusses the TDVP for infinite MPS.

  • I am honestly puzzled by "For times beyond the convergence time we also observe a strong influence ...": are data of any meaning beyond that point? Maybe it is a mismatch in what I understood, but the term "convergence time" is not defined anywhere...

We removed this discussion in the revised version for clarity. Indeed, we argue that data is meaningless past the “convergence time”, for which the threshold needs to be chosen appropriately. It is so a) because it cannot represent the exact wavefunction’s entanglement properly and b) because of the discussed issues of the TDVP with rank-deficient initial conditions as well as the non-linearity of the TDVP equations of motion. The latter is a problem if the method is supposed to be used. - What do the Authors exactly intend with dynamics of "critical one-dim. systems"? What is going to vary and what is quenched or tuned?

We intended to perform a quench to a critical point, or equivalently the dynamics of some non-equilibrium initial condition at a critical point. This point has been clarified in the discussion. - By the way, there are good reasons for a stable time-evolution algorithm be missing for MERAs (though notice Phys. Rev. A 77, 052328 (2008)), given that their structure is full of loops and already extremely hard to manipulate for static optimisations...

We agree that these are the reasons for which such algorithms are not (yet) available.

Responses to Referee 2

The manuscripts propose to use tree tensor networks to perform the time evolution of a 2D quantum system. There could be several good reasons to follow such a procedure but very few of them are carefully reported.

Weaknesses

The main weakness is

1) the lack of a critical analysis of their results,

2) the lack of an accurate benchmark of their algorithm (against i.e. ground-state physics)

In the revised version, we report absolute data for certain observables in addition to averaged deviations from the exact/reference result, which provides context for interpretation of the latter. In addition to an exact solution, which was present in the previous version of the manuscript, we add a comparison to a numerically exact solution using exact diagonalization of a small lattice, and a comparison to literature results obtained by a different method.

3) the consideration of only single-site operators

While multi-site operators are certainly also interesting quantities to provide benchmarks for, we think that the restriction to single-site operators is justified. Single-site operators are physically relevant observables, they are commonly studied objects in non-equilibrium simulations and they are frequently used indicators of the convergence of TNS algorithms.

4) the lack of description of the physics of the models considered (e.g. location of the localization transition etc)

The critical dimension for the Anderson localization is d=3, thus non-interacting spinless fermions are localized at any finite disorder strength in two dimensions. We clarified this point in the revised version.

We appreciate the referee’s critical reading of the manuscript, which allowed us to improve upon the presentation of our results considerably.

Report

I have read the paper with interest and cannot recommend it for publication in its present form.

The authors implement the TDVP for tree tensor network (something that was already done in the literature) and use it to time evolve 2D states.

They constantly mention exact results, but their benchmarks are incomplete, show a strong dependence on the bond dimension of the TTN and there is no criterium the reader can use to establish when the results are reliable.

The dependence on bond dimension is negligible at early times and becomes important only at intermediate to late times (on the timescales studied) in both models studied. In fact, the dependence of the observable of interest on the bond dimension is the criterion that we use to determine the convergence of the method in case when no exact data is available. In the revised version, we supplement our previous discussion of average deviations with absolute data for selected observables. The latter is chosen representatively and motivates the choice of convergence thresholds, and qualitatively agrees with the analysis of average deviations.

As a side remark they miss several key references on the topic (including those where the TDVP has been used to optimize 2D TTN), and seem to be completely unaware of the current trends in the TN community where people try to understand how to establish when the results of approximate time evolution are reliable. On this specific aspect, I suggest they read the introduction and references in our recent paper J. Surace, M. Piani, and L. Tagliacozzo Phys. Rev. B 99, 235115 – Published 7 June 2019 where we have tried to give an overview of the effort in the community.

We thank the referee for pointing out these relevant efforts in the community, and we are now citing them in the introduction.

In the following I list the concerns that lead me to formulate the above opinion in detail starting from the analysis of the results presented.

**Regarding the results,

The authors start studying an exactly solvable case of free fermions with disorder.

I am quite puzzled that there is no plot of the exact evolution but only of the error with respect to the exact result.

This does not provide a good insight on how large or small the error is (since it is not even the relative error they plot).

1) They should add the plot of the exact results (even if boring) and superimpose the results of their simulations on the top of it, at least in a inset of their current plots.

We appreciate the constructive criticism on the presentation of our data and are now reporting cuts of the fermion density at several times in addition to the average deviation from the exact fermion density.

2) What convergence within two means in the following sentence?

" Without disorder, convergence of the local density within 2 is obtained for both quaternary and binary TTNS only for t ≤ 1.

We missed a ‘%’ there, and corrected this typo.

They decide to only characterize local expectation values of the fermionic occupation (something pretty restrictive given that they have access to the full many-body wave function). Anyway there is a complete lack of discussion about what are the expected results for such local observable. I would assume that at least for weak disorder they should equilibrate to a value given by the Gaussian Diagonal Ensemble constructed from their initial state.

3) What is the equilibration time?

4) Is it shorter than the recurrences?

5) When the disorder is ramped up, they enter an Anderson localized phase, what is the critical strength for the transition?

6) What is the effect of such a transition in their evolution? Could they observe it?

While other, for example non-local, observables are certainly a possibility, the fermionic occupation is a valid and relevant quantity to assess the performance of the algorithm. We do not discuss the asymptotic behaviour for the fermionic occupations of this integrable system in detail due to the very limited accessible timescales. We, however, improved the discussion of the physics of the model.

7) The values of the used bond dimensions are far too small (in the binary tree they can be pushed to several >hundred/ a thousand), they should repeat the simulations with larger bond dimension until lines for the errors >with different bond dimension superimpose and then they can attribute all the remaining deviation from the >exact evolution to the Trotterization errors.

We agree that a more heavily optimized implementation, including the use of symmetries of the Hamiltonian, would certainly allow the use of higher bond dimension and yield converged results for marginally longer times. However, the range of the employed bond dimensions is sufficient to assess the performance of the algorithm and analyse its convergence properties. With respect to Trotterization errors, we provide evidence that for the chosen timesteps, they are only relevant (compared to errors related to finite bond dimension) at very early times. In addition, we provide evidence that the results agree (between different tensor network structures and bond dimensions) up to a specified and reasonably small accuracy up to the times for which the results are claimed to be converged.

8) Given the availability of exact results in this context, why haven't the authors tried to extrapolate the finite bond dimension results and see if they are able to obtain the exact ones?

9) I am also concerned with the lack of fundamental checks on their evolution. For example the evolution is unitary and should conserve the energy of the initial state. Actually the deviation of the conservation of the energy has been widely used as a measure on how good an approximate evolution is.

In the present version there are no plots about the energy conservation, and the lack of discussion about the use or not of a symplectic integrator and if and for how long the energy is conserved.

In the revised version of our manuscript we show that our results coincide with the exact results, in a way that is indistinguishable by the naked eye, up to some finite time. Therefore while is more correct theoretically to extrapolate the finite bond-dimension results (assuming it reached its asymptotic scaling), it doesn’t buy much practically.

The single-site TDVP in the implementation used here preserves unitarity by construction, and we verified that our implementation does indeed conserve norm and energy up to machine precision. We didn’t include a proof in the manuscript, but it is established for the corresponding MPS version of the integrator used here [49], and also explicitly shown for Tucker Tensors (Trees with a single layer) [59], which is at the basis of the generalization to TTNS presented here. This also holds for the particle number and the energy, but not for spatial symmetries (at least for the TTNS chosen here). Thus, we use the expectation value of local observables like the fermionic/bosonic density as checks for the validity of the time-evolution.

[49] Jutho Haegeman, Christian Lubich, Ivan Oseledets, Bart Vandereycken, and Frank Verstraete, Phys. Rev. B 94, 165116 – Published 10 October 2016 [59] C. Lubich, Applied Mathematics Research eXpress 2015, 311 (2014)

10) As a result the plots presented are not really conclusive, they show that if you run the tdvp on tree tensor network you get some results out of it. But what these results mean is far from obvious, at least for me. How does this approach compare with say a simple spin-wave analysis or semiclassical approaches like the truncated Wigner methods?

Since the method we analyze in our work is numerically exact, we do not think it is helpful to provide comparison with approximate methods Instead we benchmark with respect to exact solutions, or other numerically exact methods. It is possible that some approximate methods will give reasonable agreement with the exact solution, for a certain range of parameters, such as weak interactions, high temperature etc. It was not the purpose of the manuscript to benchmark such approximate methods. We hope that the revised manuscript with the addition of absolute data convinces the referee of the numerical exactness of the method, with the usual restriction to short times shared by all TNS methods.

11) Using the Jordan Wigner transformation for simulating fermions is definitely a possibility but I would suggest at least to mention all the theoretical development about fermionic tensor networks.

The relevant developments are now referenced in the manuscript.

12) The sentence ."..The application of TDVP formally requires the TTNS corresponding to the initial condition to possess a full tree rank of r." is formally wrong. The fact that TDVP works on subspaces with fixed bond dimension has been addressed already in the literature by passing to a two sites algorithm (and some initialization stage). I believe that adding noise to the initial state in order to fill the rank is a pretty dangerous strategy, that could lead to unexpected results My belive seems to be confirmed by the sentence "For times beyond the convergence time we also observe a strong influence on the initialization procedure of the redundant parameters of the TTNS. Increasing the bond dimension systematically reduces this effect."

13) What is the convergence time they talk about?

The TDVP does require a smooth manifold which requires the state to be of full TTNS-rank, see for example [65]. We added a discussion of all (to the best of our knowledge) approaches to fix this issue in our remarks on the algorithm. The two-site TDVP algorithm only solves the problem for local interactions in one dimension when applied to MPS (since there is no projection error). Here, the existence of local interactions in two spatial dimensions and of Jordan-Wigner strings (in the fermionic system) induces highly non-local interactions on the tree TNS, which will result in similar performance of one-site and two-site TDVP, see also the discussion and numerical experiments in [68]. We provide evidence that, at least for the models shown, the strategy of filling up the initial tensors with zeros (or a weak noise) works reliably.

With convergence time we refer to the time up to which the best and second best bond dimension results agree for a given observable up to a given accuracy. We neglected a careful definition of these two parameters in the previous version of the manuscript, and rephrased the revision such that the term convergence time is not used as such.

[65] André Uschmajew, Bart Vandereycken, The geometry of algorithms using hierarchical tensors, Linear Algebra and its Applications, Volume 439, Issue 1, 2013, Pages 133-166, https://doi.org/10.1016/j.laa.2013.03.016. [68] 5 M. Yang and S. R. White, “Time dependent variational principle with ancillary krylov subspace,” (2020), arXiv:2005.06104 [cond-mat.str-el]

All the above observations are even more relevant once they study the non-solvable system of hard-core bosons. Here we are completely lost and do not know how to address the results.

14) The quantity rho discussed in the text is defined in a label of fig. 5 with a site index (already pretty strange, I would have expected a definition in the main text). Furthermore in the definition of the anisotropy A \rho is defined with two indices that are summed over (so they supposedly represent the two coordinates of a point). However it is not clear to me where the zero of coordinates is and thus which asymmetry A is measuring.

I guess it is the crossing of the two lines in fig 5 but the reader should not guess.

We changed our notation and terminology to the occupation number (bosonic and fermionic) and added limits to the summation in the definition of the asymmetry A in the main text. (The zero of coordinates is in a corner of the lattice).

15) Furthermore there is no explicit indication about how physical sites in the lattice are mapped to the TTN, since one could do it in infinitely many ways. I guess that the obvious choice has been made, but it would be important to understand if the 0,0 point is in between the tensors or in the center of a 4 leg tensor, in order to understand better the role of the structure of the network on this geometry.

We included illustrations explaining the mapping of physical sites to the TTNS for both quaternary and binary TTNS.

16) I am very puzzled by the sentence "However, given the systematic improvement in the convergence with respect to the bond dimension combined with reduction of the anisotropy of the results, we consider the latter to achieve numerically exact results for t < 2."

Their plots show huge variations between TTN with different bond dimensions and between TTN with different structures, what is exact from their point of view?

We tightened our criterion for convergence, motivated by the analysis of the evolution of the occupation of a representative site. We are now claiming convergence up to t<1.5, for the best available data, which is of comparable quality to the MPS data of Ref [20] also shown for comparison in the Fig. 10.

[20] M. P. Zaletel, R. S. K. Mong, C. Karrasch, J. E. Moore and F. Pollmann, Phys Rev B 91, 1 (2015)

***General mistakes and lack of references

First of all there are several errors in the text.

1) In the introduction they claim "the logarithm of the latter quantity gives an upper bound on the entanglement entropy for every bipartition of the lattice" this is simply wrong and should be reformulated correctly. (Think of a bipartition in an MPS when one separates even from odd sites).

We thank the reviewer for pointing this inaccuracy out which helped make the relevant sentence precise.

2) Saying that PEPS algorithms involve uncontrolled approximations seems to me a bit too strong. But this is actually an opinion.

We made the statement less strong.

3) There is a typo "however describing to higher spatial dimension"

We replaced describing by going.

4) In the list of reference about MPS with long-range interactions key references are missing, in particular, the foundational papers are as far as I am aware

Gregory M. Crosswhite, A. C. Doherty, and Guifré Vidal

Phys. Rev. B 78, 035116 – Published 14 July 2008

Fröwis, F.;Nebendahl, V.;Dür, W.

Physical Review A, vol. 81, Issue 6, id. 062337

5) I have co-authored two papers in this context that deal with 1) the TDVP for systems with long-range interaction, in a version where the MPS is interpreted as a TTN (as you can appreciate in the drawings in the supplementary material). The algorithm is characterized both at equilibrium in Koffel, Lewenstein, Tagliacozzo, Physical Review Letters, vol. 109, Issue 26, id. 267203 and out of equilibrium in Hauke Tagliacozzo Physical Review Letters, vol. 111, Issue 20, id. 207202.

In the supplementary material of both papers we have introduced a version of the single-site TDVP these two papers should be cited in conjunction with References 16 and 36 since they both were published before the two cited papers.

6) The TDVP for studying ground states of 2D TTN was implemented and benchmarked in Andrew J. Ferris Phys. Rev. B 87, 125139 – Published 25 March 2013 the paper should also be referenced. 7) As a matter of fact, I strongly recommend using the TDVP they implement to first find the ground state of 2D systems described with TTNs (by just moving to imaginary time) and compare the results they obtain with the one available in the literature about bond-dimensions and precision (e.g. the above paper is a good starting point but they could also compare with the results of their reference 23).

The references were added to the introduction. As discussed above, the real-time evolution of local observables is a sufficiently stringent test of the algorithm’s stability and numerical exactness.

Once they are sure that their algorithm does not contain any error, they can use it to time evolve the systems and perform accurate comparisons with free systems (as indicated in the first part of this report) and then present the results for the interacting cases.

Alternatively, if they do not want to compare their algorithm with known results at equilibrium, they should at least compare it carefully with the exact results, where available, and run extensive tests against exact diagonalization results on small lattices where the analytical results are not available.

An explicit comparison between ED and TDVP for a small lattice was added to the results section.

Responses to Referee 3

Strengths

1- The manuscript is overall well written and and the results are critically discussed,

2- Finding efficient methods to simulate the real time evolution of D>1 dimensional systems is a timely and important challenge.

Weaknesses

1- Lack of references (many of them have already been mentioned by the other two referees). Also the isometric form had been used early on D. Nagaj, E. Farhi, J. Goldstone, P. Shor, and I. Sylvester, Phys. Rev. B 77, 214431 (2008).

We added the reference to the introduction.

2- From the introduction it is not getting completely clear why we should expect tree-tensor networks to perform better than matrix-product states. In particular, they have the same 1D area law restrictions.

While indeed the same area law restrictions apply, TTNS have been shown to provide a more robust approximation to the ground state of critical one dimensional systems than MPS, which is linked to the maximal distance on the tree between physical sites growing only logarithmically in the system size instead of linearly for MPS [26,27]. TTNS therefore might provide better flexibility in capturing more complicated entanglement structure in higher dimensions. We have clarified this point in the introduction.

[26] M. Rizzi, S. Montangero, P. Silvi, V. Giovannetti, andR. Fazio, New Journal of Physics12, 075018 (2010) [27] Z.-L. Tsai, P. Chen, and Y.-C. Lin, The European Phys-ical Journal B93, 1 (2020)

3- Given that the manuscript promises to provide a benchmark, a more quantitative comparison would be useful.

In the revised version, we perform a more precise convergence analysis, including a comparison of absolute data and a more stringent convergence criterion for the interacting system. The error measures and comparisons were chosen from the point of view of likely applications of the algorithm, but as stated in the conclusion, further investigation is necessary to obtain a more complete and quantitative picture of the performance of TTNS for two dimensional systems.

Report

In my opinion, the work represents a useful exploration of tree-tensor networks as a tool to study the real time dynamics. While I found the results not too surprising, it is still useful to have a benchmark of the method.

Requested changes

The existing reports already include all my criticisms, and thus I have no additional requests.

---

## Round 3 · List of Changes

• improved discussion of the results/benchmarks by 1) reporting of absolute data for a representative observables for both the non-interacting and interacting system 2) comparison of exact diagonalization and TTNS results for a small lattice in the interacting system 3) discussion of the influence of the timestep on the accuracy 4) use of a tighter convergence criterion and inclusion of reference data from published MPS results for the interacting case
  • added illustrations showing the mapping of the physical lattice to the n-ary TTNS and improved the graphical representation of central quantities in the derivation of the algorithm (Fig. 5)
  • clarified remarks on the integrator and the TDVP for TNS in general
  • added missing references suggested by referees
  • made some statements in the introduction more precise

---

## Editorial Decision

published